# LEARNING TO REASON IN LARGE THEORIES WITHOUT IMITATION

## ABSTRACT

In this paper, we demonstrate how to do automated higher-order logic theorem proving in the presence of a large knowledge base of potential premises without learning from human proofs. We augment the exploration of premises based on a simple tf-idf (term frequency-inverse document frequency) based lookup in a deep reinforcement learning scenario. Our experiments show that our theorem prover trained with this exploration mechanism but no human proofs, dubbed DeepHOL Zero, outperforms provers that are trained only on human proofs. It approaches the performance of a prover trained by a combination of imitation and reinforcement learning. We perform multiple experiments to understand the importance of the underlying assumptions that make our exploration approach work, thus explaining our design choices.

## 1 INTRODUCTION

Theorem proving is a challenging benchmark for automated reasoning, and is an important milestone on the road to demonstrating that machine learning can produce a deep understanding of abstract concepts. In the long run, automated mathematical reasoning may become an important tool in engineering and scientific discovery. Due to their success in many other areas, neural networks have recently been considered as a way to guide theorem proving (Alemi et al., 2016; Loos et al., 2017; Huang et al., 2019; Bansal et al., 2019; Paliwal et al., 2020) and demonstrate approximate mathematical reasoning abilities in latent space (Lee et al., 2020).

While there is only a relatively small number of fundamental proof rules (or proof tactics) applicable at any point in a proof, there is a very large number of premises (i.e., previously proven theorems and lemmas) that could be invoked. The largest formalized libraries have over tens of thousands of theorems that can be used as premises. Thus, the main problem of *reasoning in large theories* is to identify the premises relevant in the current context and thereby reduce the branching factor of the proof search to a manageable size. This problem will become even more pronounced over time, as the theorem provers become more powerful, growing the number of available premises.

Previous works have relied on human proofs to either directly provide or learn (Bansal et al., 2019; Paliwal et al., 2020) which premises are relevant to the current proof. However, any open-ended system for mathematical reasoning needs to be able to learn which premises are relevant without human guidance. In this work, we thus consider the problem of training a theorem prover *without access to human proofs*. In particular, the contributions of this work are:

1. We demonstrate training the theorem prover without human data can succeed when using deep reinforcement learning. We do this with minimal additional engineering: by augmenting exploration of premises with a portion of the premises selected by a tf-idf (Manning et al., 2008) metric.

2. We provide a first side-by-side comparison of the effect of availability of human proofs on the final theorem proving performance. We learn to prove more theorems than the prover trained on human proofs alone and almost as many as with the combination of both approaches.

3. We establish the underlying properties of the proof assistant and reinforcement learning setup that makes our approach work, by running multiple ablation experiments.

We thereby solve one of the road blocks on the way to open-ended learning of mathematical reasoning in large theories.

## 2    RELATED WORK

Reinforcement learning (RL) without imitation learning has been successful for computer games (cf. Mnih et al. (2013)) and it was demonstrated later in Silver et al. (2017) that imitation learning is not necessary for complex games like Chess and Go. For more complex games with much larger action spaces, learning methods still rely on human imitation due to the exploration problem (cf. Vinyals et al. (2019)). The question of exploration is well studied in reinforcement learning (Houthooft et al., 2016; Burda et al., 2019), but existing approaches such as $\epsilon$-greedy do not work for premise selection because of very large (practically infinite) action space.

We work in the setting of automating higher-order logic interactive theorem provers, since this is where there is most promise for building and formalizing large theories. This is also evidenced by the fact that all large-scale formalization efforts by mathematicians have occurred in such systems (Gonthier, 2008; Hales et al., 2017). Several works have explored RL for proof search in the context of connection provers (Färber et al., 2017; Kaliszyk et al., 2018; Zombori et al., 2019; 2020). We are instead interested in addressing the issue of premise selection from a large knowledge base, through the use of deep reinforcement learning and without use of human proofs. This is the hard part of exploration due to the large repository of premises.

Premise selection itself has been an active research topic in the domain of automated theorem proving (Alama et al., 2014; Kaliszyk and Urban, 2015; Blanchette et al., 2016; Wang et al., 2017). Gauthier et al. (2017) uses a tf-idf based premise selection model, but does not learn a model. Urban et al. (2008); Kaliszyk et al. (2014); Kaliszyk and Urban (2014); Piotrowski and Urban (2018) interleave runs of an automated theorem prover and a premise selection model using non-deep RL approaches. Deep learning has since significantly improved the state of the art for premise selection, starting with Alemi et al. (2016), but these approaches have relied on human proofs. In our work, we use deep RL to learn premise selection while removing this dependence on human proofs. We also provide a clear comparison of the effect of availability of human proofs to final theorem proving performance, which has been lacking in the literature.

We use the HOList environment (Bansal et al., 2019) for HOL Light (Harrison, 1996). Other ML environments for proof assistants include GamePad (Huang et al., 2019) and CoqGym (Yang and Deng, 2019) for Coq; and TacticToe (Gauthier et al., 2017) for HOL4 (Slind and Norrish, 2008).

## 3    BACKGROUND

**Theorem proving.**    Proof assistants have been built to enable humans to write and then automatically check proofs. In contrast to mathematical textbooks and papers, which are written mostly in natural language, we call mathematics formalized in proof assistants to be *formal mathematics*. In this work we focus on the proof assistant HOL Light (Harrison, 1996), in which a wide range of mathematical theories have been formalized, and which has been famously used for the formalization of the Kepler conjecture (Hales et al., 2017). HOL Light, as in many other proof assistants, relies mostly on "backward" proof steps. In contrast to "forward" proof steps, in which we only manipulate already proven statements, backward proofs start with a proof goal (the statement of the theorem to be proven) and apply proof tactics until all goals are proven.

In Figure 1, we give an example of a backward proof. The goal here is to prove $x + 0 = x$, for all $x \in \mathbb{N}$, and we apply the tactic MATCH_MP_TAC to the goal. Like many tactics, this tactic takes a

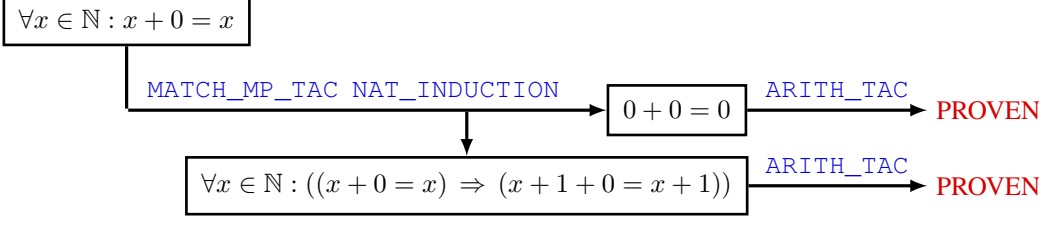

Figure 1: Formally proving $\forall x \in \mathbb{N} : x + 0 = x$.

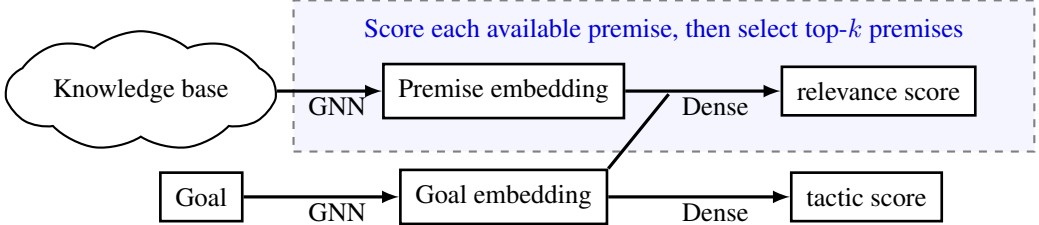

Figure 2: Architecture for tactic and premise selection by Paliwal et al. (2020). Note that this work is largely agnostic to the model architectures.

*premise* (i.e. a previously proven theorem or lemma) as a parameter. In this example, we use the induction theorem `NAT_INDUCTION` as a premise. This tactic application splits the first goal into two subgoals, corresponding to the base case and the induction step. The semantics of an application of a proof tactic is that, if all subgoals are proven, then also the goal to which the tactic has been applied is proven. In our case, we can prove both of the subgoals by simple arithmetic reasoning, provided by the tactic `ARITH_TAC`. This tactic here does not require additional premises and returns an empty list of subgoals (for both of the subgoals we apply it to), meaning that they are proven, and hence the original goal is proven.

**Learning proof guidance for interactive theorem proving.** It is a long-standing goal in artificial intelligence to automate the theorem proving process described above, in particular to relieve the human experts from selecting the tactics and premises in each proof step. Historically, most works focused on designing advanced search algorithms, leading to entire fields such as SAT and SMT solving and first-order theorem proving. Recently, learning proof search strategies from data has become an area of active research (Alemi et al., 2016; Gauthier et al., 2017; Huang et al., 2019).

In this work, we follow the approach by Bansal et al. (2019), which has shown the unique ability to find relevant premises, which has been a big challenge for the classical techniques. Figure 2 illustrates the tactic and premise selection architecture introduced by Bansal et al. (2019) and improved by Paliwal et al. (2020). For each proof step, this architecture scores the tactics and it also produces a relevance score for each potential premise. Then, for each tactic, the top-$k$ premises are given as arguments (unless the tactic does not require premise arguments). This results in a list of candidate tactic applications, which can be used as the actions in any search approach. We adopted the same search strategy as Bansal et al. (2019) and Paliwal et al. (2020), which is a simple breadth-first search with a parameter of how many of the candidate tactic applications should be expanded per proof goal.

The tactic and premise selection architecture is trained on successful proofs. For imitation learning, tuples of goal, tactic, and used premises are extracted from human proofs formalized in HOL Light. The focus of this work, however, is to not learn from human proofs, and instead learn in a reinforcement learning setup from the proofs that earlier versions of the policy have found.

## 4    LEARNING WITHOUT IMITATION

In this section, we explain the setup for learning to prove in the absence of human proofs, and the considerations that informed our final design.

Much of mathematics that has been formalized by humans is in pursuit of formalizing certain theorems such as the four color theorem (Gonthier, 2008) and the Kepler conjecture (Hales et al., 2017). Since formalization is a challenging and tedious process requiring experts, a very small fraction of mathematics is formalized after decades of human effort. This work paves the way for a critical piece of a system wherein its knowledge base of formally proven theorems grows continuously. We separate two key aspects of such a system. First, proposing potentially true statements, also known in the literature as conjecturing. Second, given a new statement, proving it without existing proofs to learn from. We would like to tackle the latter question in this work directly.

The key information the human proofs provide is the overall direction of the proof via selection of the relevant premises at each proof step. In case human proofs are available, one can first train a machine learning model to imitate them (Section 3), as has been done in several previous works (Section 2).

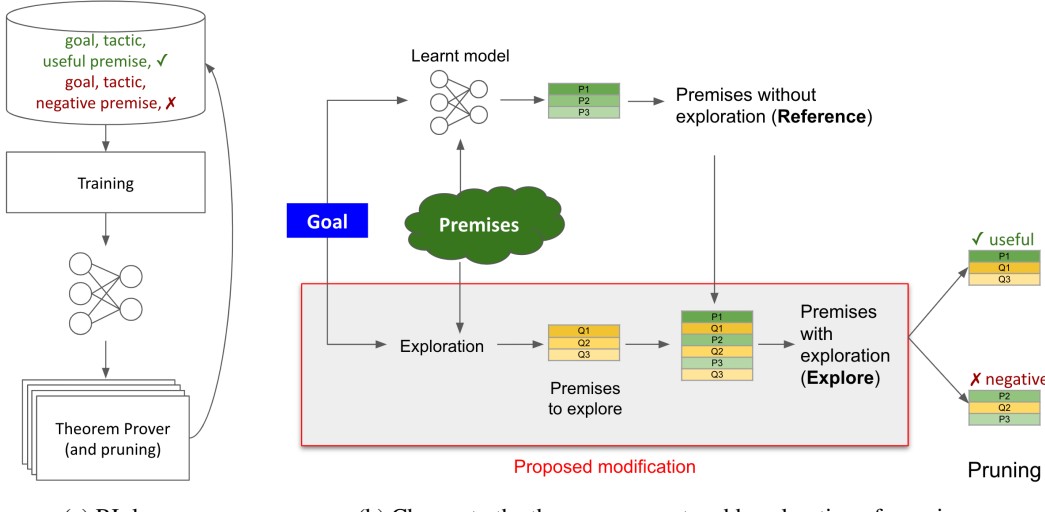

(a) RL loop

(b) Change to the theorem prover to add exploration of premises.

Figure 3: The figure on the left gives a high-level overview of the components in the reinforcement learning (RL) loop. The figure on the right shows how the model being trained is used for premise selection. We propose a modification to the premise selection process to aid exploration in the RL loop.

**Reinforcement learning loop.** In the absence of human proofs, we need a mechanism to incrementally improve a proof guidance model, which motivates the reinforcement learning setup we use. Figure 3a shows the components of the reinforcement learning loop from Bansal et al. (2019), which we build upon. A proof guidance model is trained with continuously expanding training data. In order to generate the continuously expanding training data, several theorem provers run in lockstep with training the policy and premise selection network. The provers try to prove the statements in the training set using the model for proof guidance as it is training. If it manages to prove a statement, the proof is used to generate additional training data. Intuitively, we would like to reward the choices of tactics and premises that were "useful" at each step in the proof. A subtle but crucial aspect is that the proofs are pruned before generating the training data. In this step, premises that are not necessary for the proof are removed. This interplay between over-approximation and pruning is a major contributing factor to the efficiency of our exploration method and is studied in Subsection 4.2.

Figure 3b shows how the list of premises are picked, including our proposed change to aid exploration of the premises. Given a goal, the currently learnt model is used to pick the top-$k_1$ highest scored premises from the knowledge base of premises available: $\{P_1, P_2, \ldots, P_{k_1}\}$. Simultaneously, we propose generating another list of premises $\{Q_1, Q_2, \ldots Q_{k_2}\}$ to explore, picked according to a metric discussed shortly (Section 4.1). The final set of premises is obtained by interleaving the two premise lists. $k_1$ and $k_2$ are hyperparameters which can be varied to control exploitation of the learnt model (higher $k_1$) vs exploration (higher $k_2$). The REFERENCE setup is one without our modification (i.e., $k_2 = 0$). On the other hand, the EXPLORE setup includes additional premises as proposed here.

There are three aspects to this that we wish to highlight, and that have informed the design: the strategy to generate the list of premises to explore, the effect of using irrelevant premises in an action (over-approximation of premise lists), and pruning. We discuss each of these in detail, designing experiments to inform different choices.

## 4.1 INFORMATION RETRIEVAL FOR PREMISE SELECTION

One of the key failure modes of our reinforcement learning setup trained without human proofs is not being able to prove new theorems. With no new training data, the learning process would stall. Thus, it is crucial that we continuously expand the horizon of theorems that we are able to prove. Since we generate new training data only when we manage to prove a theorem, we wish to pick premises most likely relevant to prove the current goal.

Table 1: Retrieval performance on human proof logs

| Term Freq. | av rel max rank | recall@16 | recall@32 | recall@64 | recall@128 |
|------------|-----------------|-----------|-----------|-----------|------------|
| boolean | **0.24** | **0.15** | **0.19** | **0.25** | **0.31** |
| logarithm | 0.35 | 0.1 | 0.13 | 0.17 | 0.21 |
| natural | 0.46 | 0.06 | 0.08 | 0.09 | 0.11 |

In information retrieval literature, notions such as tf-idf (Manning et al., 2008) have been used to retrieve relevant documents corresponding to a query. We view premise selection as a retrieval problem, thinking of the current goal we are trying to prove as the query, and the knowledge base of previously proven theorems (premises) as the documents from which we would like to retrieve.

Given a goal $G$, we use pre-engineered similarity scoring $s(G, P)$ to rank a potential premise $P$ for its usefulness in the next action (tactic application) with the target of proving the goal. In our setup, we restrict our attention to functions of the form $s(G, P) = \langle r(G)/\|r(G)\|, r(P)/\|r(P)\| \rangle$, where $r$ is some simple vector representation of the expression and $\langle \cdot, \cdot \rangle$ the dot product. This is also sometimes referred to as the cosine simarity. We consider $r$ of the form $r(P)_i = \text{tf}(P, i)\text{idf}(i)$, where the $i$-component corresponds to the tokens occurring in the formulas, idf is the "inverse document frequency" function which is precomputed by $\text{idf}(i) = \log(N/n_i)$, where $N$ is the total number of theorems and $n_i$ is the number of theorem containing the $i$-th token. For the term frequency function $\text{tf}(P, i)$, we have tested three possible choices: boolean weighting: 1 if $f_i(P) > 0$ and 0 otherwise, logarithm weighting: $1 + \log(f_i(P))$ and natural: $\text{tf}(P, i) = f_i(P)$, where $f_i(P)$ is the number of occurrences of the $i$-th token in expression $P$. The number of tokens in our experiments was 885.

Running a full reinforcement learning loop uses over 25 years of CPU resources (Appendix B.2). Since it is prohibitive to run a lot of experiments with the full system, it suggests that we should first evaluate the quality of similarity metrics in an offline manner. That is, we measure how well those metrics perform on existing proof traces and we only apply the similarity scores that performed best in a separate evaluation. To do so, we have adopted the metrics from Alemi et al. (2016). We have measured the average relative maximum rank and the top-$k$ recall numbers for a small set of relevant $k$ values (for $k = 8, 16, 32$ and $64$) on a random selection of proofs comprising of $20\%$ of the training set of the "complex" corpus of HOList. The relative maximum rank of a true positive document (premise) is the absolute maximum rank of the true documents divided by the size of all possible premises (from which we make the selection), the average is then taken over all retrieval tasks. The results are summarized in Table 1. Note that low average maximum relative rank and high recall values indicate better premise selection performance of the similarity measure.

To summarize, we pick the **boolean** term-frequency weighting scheme. In addition, we speculate it could be helpful to add more variation of premises picked for exploration for a given goal, as over the course of a loop same goals are attempted multiple times. To add this variation, we add a dropout probability hyperparameter $p$ to components of the representation: $r(G)_i$ is zeroed with probability $p$ when computing the representation for $G$. $p$ is set to $0.1$ unless specified otherwise.

## 4.2 OVER-APPROXIMATION OF PREMISES

Our exploration is based on the assumption that adding a few irrelevant premises does not influence the outcome of tactic applications significantly. This allows us to accelerate the search by trying to over-approximate the set of premises used in tactic applications. We study the behavior of the most frequently occurring tactics that take premises as an argument: `MESON` and `REWRITE`.

`MESON` is based on a first-order logic solver. We study how many extra premises we can add before `MESON` starts to fail to prove a goal. To do so, we first sample at random proof steps from the human proofs (which are always successful). For each proof step we add as tactic arguments random irrelevant premises from the knowledge base of premises available at that proof step. We report the ratio of successful proof attempts after adding a set of tactic parameters with varying cardinality. For `REWRITE` operations, we study the number of extra premises we can add and expect not to change the outcome of the rewrite operation. For both experiments, we sampled random proofs and one random proof step with the desired type of tactic (`MESON` or `REWRITE`), until we sampled 250 steps successfully. Then for each $l \in \{1, 2, 4, 8, 16, 32\}$, we sampled five different random parameter lists with length $l$. In our experiment, we append those parameters to our parameter list and execute the

Table 2: Tactic success rates with extra random parameters, 1 second timeout.

| Number of extra premises | 1 | 2 | 4 | 8 | 16 | 32 |
|---|---|---|---|---|---|---|
| `MESON` success rate | 0.995 | 0.986 | 0.97 | 0.873 | 0.53 | 0.06 |
| `REWRITE` unchanged rate | 0.99 | 0.979 | 0.954 | 0.93 | 0.858 | 0.731 |

same tactic with the extended parameter list. Table 2 shows the ratio of application with the outcome being identical with that of the tactic application without the extra parameters. We can see that even adding 32 extra random parameters does not change the outcome of the rewrite tactics over 70% of the time. However, `MESON` tends to time out with more than 32 extra premises.

### 4.3 PRUNING

As discussed in Section 4.2, if a tactic application succeeds, not all premises provided might have been used. In fact, we are using this fact to accelerate our exploration. However, we do not wish to learn from these irrelevant premises. Thus, for generating training data, for each proof step in our proof we greedily try to remove the premises: given a proof step with premises $\{P_i\}_{i=1}^n$, we rerun the proof step without $P_n$. If the result of the proof step remains unchanged, we drop $P_n$. Then, we continue with trying to drop $P_{n-1}$, and so on. We use the dropped premises as hard negatives, such that premises which are ranked highly by the model but are not useful are demoted. Demoting pruned premises allows other premises that had a high score but did not make it into the top-$k$ to get a chance in the future. Pruning also ensures that any extra premises added for exploration that are in fact unnecessary are not learnt upon.

## 5 EVALUATION

**Environment and benchmark.** We evaluate our approach in the HOList environment (Bansal et al., 2019) based on the HOL Light proof assistant. We chose to use HOList because of the breadth of topics of mathematics in the dataset. Additionally, HOList is already integrated into a reinforcement learning setup, which our approach relies on. We conduct our experiments on the "complex" corpus of the HOList benchmark derived from theorems in HOL Light's mathematical library from various areas of mathematics such as topology, multivariate calculus, real and complex analysis, geometric algebra, and measure theory. It includes well-known theorems such as Abel's theorem for power series, the fundamental theorem of calculus, and that the roots of the characteristic polynomial of a complex matrix are its eigenvalues. Table 3 gives some addtional examples.

The task is to attempt to prove a theorem in the benchmark, with theorems and definitions appearing before it in the benchmark available as premises. The evaluation criterion is the fraction of theorems proven with this constraint. The benchmark comes with theorems divided into training, validation, and test sets. Table 4 gives the statistics of theorems in the "complex" corpus of the benchmark. Definitions (totaling 637) and "core" corpus of theorems (totaling 2320), containing basic mathematics which the "complex" corpus builds upon, are available as premises, but are not attempted to be proven.

Table 3: Examples of theorems in the benchmark (compressed for brevity)

| Alternative characterization of orthogonal matrices. |
|---|
| $\forall A. orth(A) \iff (\forall i. \|A_i\| = 1 \land \forall i! = j. A_i \perp A_j)$ |
| Property about absolute neighborhood retract (ANR). |
| $\forall S \subseteq R^n. \text{ANR}(\text{frontier}(S)) \implies \text{ANR}(\text{closure}(S))$ |

Table 4: Benchmark statistics

| Split | # of Theorems |
|---|---|
| Training | 10214 |
| Validation | 3225 |
| Testing | 3184 |
| Total | 16623 |

**Training and evaluation.** During training, we generate data for training by trying to prove statements in the training set of 10,214 theorems. We train for 8 million steps. Details of our hardware setup and hyperparameters are in the Appendix. For evaluation of all our experiments trained in the reinforcement learning setup, we focus on the number of statements proven in the held-out validation set of 3,225 theorems. We run a continuous evaluation on samples of the validation set as well as a final evaluation on the full validation set. These metrics are:

| | Final validation | Cumulative validation |
|---|---|---|
| *Pure human imitation* | | |
| Bansal et al. (2019) | 32.65% | - |
| Paliwal et al. (2020) | **49.95**% | - |
| *Human RL* | | |
| Bansal et al. (2019) | 38.9% | not reported |
| Human reference | 59.5% | 68.2% |
| Human explore | **59.9**% | 69.1% |
| *Zero RL* | | |
| Zero reference | 7.0% | 7.3% |
| Zero explore | **56.3**% | 64.2% |

Figure 4: Results of our main experiment. We report the percentage of validation theorems proven on the HOList benchmark. The numbers in bold are the state-of-the-art in their respective categories (including results from this work). The main takeaway is that the best RL loop trained without human proof data outperforms the model trained purely on human data, and approaches the performance of the best RL loop trained with human data.

- **Continuous** validation performance (represented by dots in the plots) runs every 80,000 training steps on a *random sample* of validation theorems, and reports the fraction of proven theorems from that sample. Since not all validation theorems are attempted and the sample changes each evaluation, the metric is slightly noisy, but it allows us to monitor overfitting during training.
- **Final** validation performance (reported in the tables and plots) is the fraction of all validation theorems proven by the final checkpoint at 8 million steps. This metric also allows for comparison with models trained purely by imitation learning.
- **Cumulative** validation performance – reported in the tables and plots – is the fraction of all validation theorems proven by any continuous-validation run up until that point in the loop. The table reports the cumulative performance of the whole loop (i.e., after 8 million steps).

**Hardware and hyperparameters.** One of hyperparameters introduced is the number of additional premises picked by the model ($k_1$) vs for additional exploration ($k_2$). Since it is computationally expensive to run a single RL loop, we do not fine-tune these. Rather than picking a fixed value in a experiment, we make attempts over the course of a single experiment with different values to add more diversity. In partcicular, $k_2$ is picked uniformily at random in the range $[0, 8]$ independently by each prover. $k_1$ is then picked as $k$ (total number of premises) minus $k_2$. For evaluation, to keep our numbers comparable we use the same prover timeout and maximum number of nodes explored as in Paliwal et al. (2020). A full discussion about the hardware, hyperparameters and computational resources used can be found in the appendix.

**Main experiment.** In the main experiment we are interested in understanding the ability of the EXPLORE approach proposed in Section 4, in particular, its ability to learn to prove without human proof data available. This experiment is referred to as ZERO EXPLORE. As we see in the plot on the right in Figure 4, the loop continuously expands the horizon of the theorems it is able to prove.

To put the performance in context, we categorize the results on this benchmark into three categories. First, *pure human imitation*, wherein the model is trained only on human proof data and no theorem prover is used during training. Second, *human RL*, wherein the the model is trained on human proof data as well data generated by running the prover in a reinforcement learning (RL) loop. Finally, *zero RL*, wherein no human data is available, and all data is generated by running a prover in an RL loop. The results are summarized in the table in Figure 4.

Paliwal et al. (2020) based on graph neural networks (GNN) is the state-of-the-art on pure human imitation on this benchmark, and we use the network architecture in our experiments as well. Compared to pure human imitation, the ZERO EXPLORE RL loop using no human proofs is able to prove more theorems on a single checkpoint: 49.95% (pure imitation) vs 56.3% (zero RL).

Next, we compare to the best human RL loop (HUMAN EXPLORE), one with everything identical as in ZERO EXPLORE, except we let the model learn from additional human proof data. We see that the

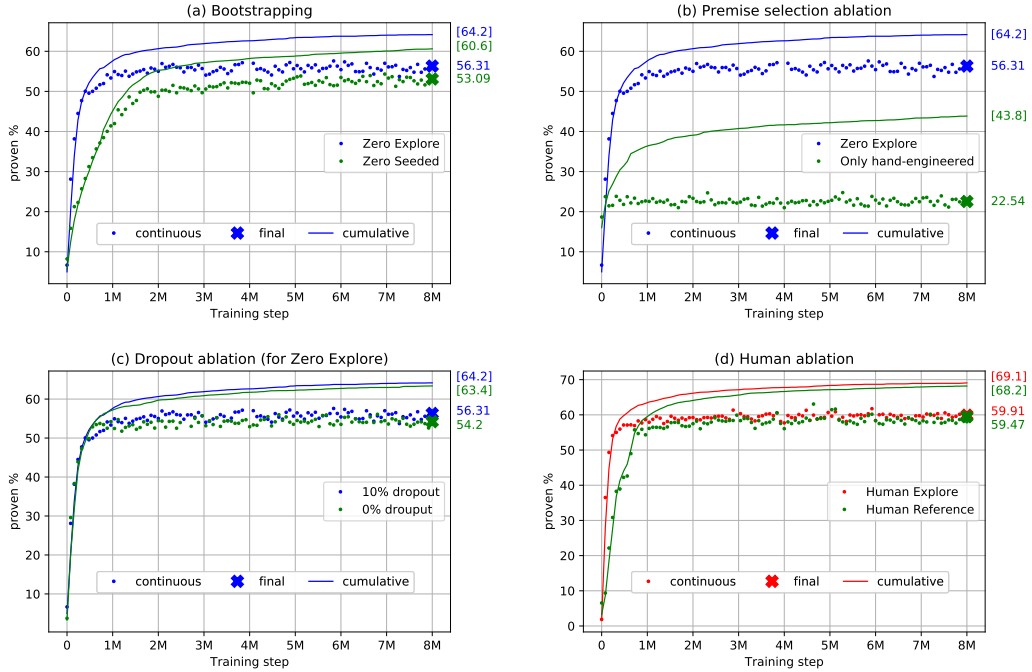

Figure 5: Ablation experiments.

ZERO EXPLORE loop comes very close to the performance of the corresponding human RL loop: 59.9% vs 56.3% on a single checkpoint, and 69.1% vs 64.1% cumulatively. ZERO EXPLORE is able to reach over 90% of the human RL loop's performance. This is an important indicator as HUMAN loops indicate a ceiling unrelated to availability of proof data: such as the proof search algorithm, or model architecture.

Finally, we compare against the RL setup from Bansal et al. (2019), and run it without human proof data (ZERO REFERENCE). We run into the failure mode of not being able to prove new statements and thus stalling, discussed in Section 4.1.

## 6   ABLATION STUDIES

**Bootstrapping.**   In Section 5, we observe that the ZERO REFERENCE loop stalls very quickly. Here, we try to understand to what extent is the failure a bootstrapping issue. We attempt to prove all statements in the training set using the best hand-engineered metric in Section 4.1 for premise selection and random tactic selection. This proves around 20% percent of the statements. We start a zero RL loop as in ZERO REFERENCE, but providing these additional proofs to train upon, calling it ZERO SEEDED. We see in Figure 5a that it does not stall like the reference loop. At the same time, it does not reach the same level of performance as the ZERO EXPLORE, which explores throughout.

**Premise selection ablation.**   Through this ablation, we wish to understand the capability of the hand-engineered metrics in Section 4.1 to prove theorems. To keep the focus on premise selection, we still learn the tactic selection, and use similar amounts of resources trying to prove statements as in one RL loop. Technically, we do this by setting $k_1$ (number of premises chosen by the learnt network) to 0, and rest of the setup as in the ZERO EXPLORE loop. In Figure 5, under premise-selection ablation we see the results: it manages to prove 43% of the statements cumulatively, compared with 64% when learning with a combination of exploration and exploitation in the RL loop.

**Dropout.**   In Section 4.1 we suggest a 10% token dropout probability when deciding premises to explore to introduce more diversity of premises in the loop overall at the cost of picking slightly less relevant premises at a specific point. We evaluate this experimentally (dropout ablation in Figure 5), but do not see a major difference: we can observe a very slight gain requiring further verification.

**Human loop ablation.**    We run a human loop where we do not add premises for exploration (human ablation, Figure 5). We do not see a significant difference in the performance. It is not surprising, as the human loop has proofs for all statements and is thus not reliant on premise exploration to find relevant premises, unlike the zero RL loops.

## 7    CONCLUSION

In this work, we demonstrate that it is possible to learn a premise selection model for theorem proving in the absence of human proofs. We show that, on our benchmark, we exceed the performance of a network trained purely on human proofs, and approach the performance of the system that combines reinforcement learning with imitation learning.

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

# A   HYPERPARAMETERS

## A.1   POLICY NETWORK TRAINING PARAMETERS

- batch size: 16 goals, 256 premises
- number of workers: 8
- optimizer: Adam
- Adam epsilon: 1e-3
- initial learning rate: 1e-4
- learning rate decay: exponential, 0.98/100000 steps
- embedding size: 128
- non-linearity: ReLU
- hidden layer dropout: 0.5
- GNN hops: 16
- layers per hop: 2
- initializer range: 0.02
- pre-combiner embedding size: 4096
- number of combiner layers: 3
- ratio of human training data: 0.7 (for human loops), 0.0 (for zero loops)
- ratio of historical training data: 0.2 (for human loops), 0.5 (for zero loops)
- ratio of fresh training data: 0.1 (for human loops), 0.5 (for zero loops)

## A.2   PROVER HYPERPARAMETERS

Some parameters are picked independently by each prover. These are picked uniformly at random from a given interval, are indicated below as $[n_1, n_2]$. The intervals are inclusive of both end points.

**Training provers:**

- total number of provers: 2000 (see continuous validation note)
- training round interval (each time a model checkpoint is written out): 4000 training steps
- prover tree search strategy: BFS
- timeout: 300 seconds
- maximum number of actions considered (per goal): [10, 30]
- maximum *successful* actions (per goal): [10, 18]
- maximum number of premises ($k$): [2, 32].
- number of premise samples per tactic: 4
- tactic timeout: 500 milliseconds
- maximum goals explored: 1000000 (practically, no limit)

Parameters that vary from experiment-to-experiment:

| Loop name | Seed proofs | | Premise selection | |
| --- | --- | --- | --- | --- |
| | Human | Generated | Learnt model | Exploration |
| Human reference | Yes | - | Yes | No ($k_2 = 0$) |
| Human explore | Yes | Exploration | Yes | Yes ($k_2' \in [0, 8]$, $p = 0.1$) |
| Zero reference | No | Reference | Yes | No ($k_2 = 0$) |
| Zero explore | No | Exploration | Yes | Yes ($k_2' \in [0, 8]$, $p = 0.1$) |
| Zero seeded | No | Exploration | Yes | No ($k_2 = 0$) |
| Zero hand-engineered only | No | Exploration | No | Yes ($k_2 = k$, $p = 0.1$) |
| Zero explore 0% dropout | No | Exploration | Yes | Yes ($k_2' \in [0, 8]$, $p = 0.0$) |

$k_1$ (number of premises to be picked by the network) and $k_2$ (number of premises to be added for exploration) are decided as follows. In the table above, for experiments where $k_2$ is defined $k_2$ is picked as shown. For experiments where $k_2'$ is defined, $k_2$ is derived it as $max(\lceil k/2 \rceil, k_2')$. $k_1$ is picked as $k - k_2$.

$p$ is the token dropout probability, defined in Section 4.1.

For technical reasons, for the training to start some non-empty training data is needed. For human RL loops, since human proof data is available, the generated seed data is not strictly necessary. For zero RL loops, some data needs to be provided. We generate this data by trying to prove all theorems in the training set on a randomly initialized model (i.e. the 0-th checkpoint of a model). The hyperparameters used for the provers to generate the seed data is as follows:

- prover tree search strategy: BFS
- timeout: 1000 seconds
- maximum number of actions considered (per goal): 20
- maximum *successful* actions (per goal): 5
- maximum number of premises: $k = k_1 = 24, k_2 = 0$ (reference), $k = k_2 = 16, k_1 = 0, p = 0.0$ (exploration).
- number of premise samples per tactic: 1
- tactic timeout: 5000 milliseconds
- maximum goals explored: 1000000 (practically, no limit)

**Validation provers:**   To keep this work comparable to prior results on HOList benchmark, we have verified that the timeout and maximum explored nodes are the same as in Paliwal et al. (2020). We have also verified that the results in Bansal et al. (2019) are within half a percent of the reported results with these timeout and maximum explored nodes.

- prover tree search strategy: BFS
- timeout: 1000 seconds
- maximum number of actions considered (per goal): 20
- maximum *successful* actions (per goal): 14
- maximum number of premises: 20
- number of premise samples per tactic: 4
- tactic timeout: 500 milliseconds
- maximum goals explored: 1000000 (practically, no limit)

Following only apply to continuous validation, not final validation which is run at final checkpoint, and on the full set.

- continuous validation interval: 20 rounds (80,000 training steps)
- probability to prove a validation theorem: 0.1 (Explanation: the 2000 provers running in the RL loop for training, each independently decides to re-purpose itself with this probability to help with continuous evaluation. With this probability, this leads to over 2000 proof attempts per validation interval on average.)

## B   COMPUTATIONAL RESOURCE ANALYSIS

### B.1   HARDWARE SETUP

We used eight NVIDIA Tesla V100 GPUs for distributed training, an additional GPU was used purely for evaluation, and we maintained a separate parameter server on a CPU machine. The provers generating training data and running validation run distributed, using 2000 CPUs.

## B.2   ABSOLUTE RESOURCE USAGE ESTIMATE

Here we estimate the resource usage of a reinforcement learning loop. Since the improvement of the premise guidance is heavily reliant on generation of data, we run up to 2000 theorem provers distributing the statements each prover is attempting to prove. Computing predictions takes a few milliseconds but actions in the proof assistant can take up to half a second. We use 8 GPUs for training the policy network and the experience collection uses CPUs only. Combined with proof search, to have a reasonable chance of proving a statement, we run the theorem prover with a timeout of 5 minutes for a statement in the training set. Training over 5 days, a single reinforcement learning loop takes over 25 years of CPU resources and 960 hours of GPU resources.

## B.3   RELATIVE RESOURCE USAGE ESTIMATES

For *pure human imitation*, there is no reinforcement learning, and thus is the least computationally intensive. The models in Paliwal et al. (2020) were trained for 1 million training steps. The best model took 26 hours to train with 8 GPUs, or around 200 GPU hours.

The resource usage for reinforcement learning loops is estimated above (Appendix B.2). The resource usage comparison of *explore* versus *reference* loops is analyzed next, being the main change proposed in the work. The main difference is an additional ranking based on tf-idf. The time to rank based on tf-idf is very small compared to the ranking with the graph neural network. Nevertheless, note that the overall timeout for proving is fixed (5 minutes), and includes time to generate the actions as well as performing the actions in the environment.

One way to compare training of these different models is to consider how long each model takes to reach a certain performance. For instance, we could try to reach performance of the best Paliwal et al. (2020) model, that of around 50% validation performance. This is a rough estimate, but the best Human RL loop (*human explore*) reaches this performance after utilizing 68 hours of GPU resources and 2 years of CPU resources. On the other hand, the best Zero RL loop (*zero explore*) utilizes roughly 1.35 times the resources – 92 hours of GPU resources and 2.6 years of CPU resources.

Alternatively, one could compare the performance after utilizing the same amount of training steps. For instance, say, after roughly 1 million steps which corresponds to roughly 120 hours of GPU resources and 3.4 years of CPU resources. At this stage, the *human explore* is able to prove roughly 58% of the statements, whereas the *zero explore* is able to prove 55% of the statements. Note that these numbers are for continuous validation at 1 million steps, so are bit approximate. At the same point the cumulative validation performance is 63.5% and 57.2% for human explore and zero explore respectively.

## C   HOL LANGUAGE FOR EXPRESSIONS

The following is not necessary to understand this paper, but given for the sake of completeness. The data in the HOList benchmark extracted on HOL Light and given in the form of 'S-expressions'. This may be thought of as a serialized version of the abstract syntax tree of *terms* in HOL Light, given in prefix notation. All terms have a well-defined *type*, which is also contained in the S-expressions.

For instance, a boolean variable, would be given as `(v bool x)`. Here `v` denotes that the term is a variable, `bool` denotes the type, and `x` denotes the name. A variable named $f$ denoting a function from boolean to boolean is represented as `(v (fun (bool) (bool)) f)`. `(a (v (fun (bool) (bool)) f) (v bool x)` denotes $f(x)$, application of the function variable $f$ to the variable $x$.

As a longer example, the second example in Table 3 about absolute neighbourhood retracts is represented as `(a (c (fun (fun (fun (cart (real) N) (bool)) (bool)) (bool)) !) (l (v (fun (cart (real) N) (bool)) s) (a (a (c (fun (bool) (fun (bool) (bool))) ==>) (a (c (fun (fun (cart (real) N) (bool)) (bool)) ANR) (a (c (fun (fun (cart (real) N) (bool)) (fun (cart (real) N) (bool))) frontier) (v (fun (cart (real) N) (bool)) s)))) (a (c (fun (fun (cart (real) N) (bool)) (bool)) ANR) (a (c (fun (fun (cart (real) N) (bool)) (fun (cart (real) N) (bool))) closure) (v (fun (cart (real) N) (bool)) s))))))`

For a more detailed, yet gentle, introduction on HOL Light, we refer the reader to its tutorial (Harrison, 2011).

