# OpenReview forum: "Learning to Reason in Large Theories without Imitation"
_ICLR.cc/2021/Conference — Reject_

### Official Review · AnonReviewer3 · 2020-10-26
**Review: Learning to Reason in Large Theories without Imitation**

**Rating:** 6
**Confidence:** 3

**Review:**

This paper aims at improving high-order theorem proving using deep reinforcement learning. The key idea of this study lies in the choice of premises for helping the prover. Here, the list of premises is formed by interleaving $k_1$ premises of highest score and $k_2$ premises to explore. The sublist of exploration premises is generated using a standard TF-IDF metric. The authors study how many extra premises can be added before the prover fails. They also propose a simple pruning technique for rapidly demoting irrelevant premises and promoting potentially relevant ones. Experiments using the HOLlist framework corroborate the interest of the approach.

Overall, this approach is conceptually simple and the experimental results look promising. Yet, I am not entirely convinced that this study provides a novel contribution since it is essentially using well-known techniques.

My main comment is that the hyper-parameters $k_1$ and $k_2$ have been carefully tuned, according to the table in Appendix A. It seems that the problem of choosing the top-k premises (included exploration premises) is intimately connected to online learning to rank with top-k feedback (e.g. Chaudhuri & Tewari, 2017). So, instead of choosing $k_1$ and $k_2$ at hand, could we use a bandit algorithm for selecting premises? Here, the “context/user” would be the theorem to prove, and the “items/documents” would be the candidate premises.

As a minor comment, it would be nice to give more details about the (higher-order) language used to define expressions. It would help the reader in understanding more precisely the components of $r(G)$ and $r(P)$.

----
Sougata Chaudhuri and Ambuj Tewari. Online learning to rank with top-k feedback. Journal of Machine Learning Research 18 (2017) 1-50

---

> ### Author Response · Authors · 2020-11-17
> **Re: AnonReviewer3**
>
> Thank you for the review, and helpful comments and questions.
>
> "not entirely convinced that this study provides a novel contribution since it is essentially using well-known techniques"
> - While our system incorporates several well-known techniques, the novelty of our work lies in how these techniques are combined, and used to overcome an important so-far unaddressed problem, namely, learning premise selection from large knowledge bases (tens of thousands of theorems) without human proofs.
>
> "My main comment is that the hyper-parameters $k_1$ and $k_2$ have been carefully tuned, according to the table in Appendix A."
> - The parameters k1 and k2 were not tuned, because it is computationally expensive to run a single RL loop (Appendix B.2). This was in the appendix, so might have been missed -- "Some parameters are picked independently by each prover. These are picked uniformly at random from a given interval, are indicated below as $[n_1, n_2]$." For $k_2$, in explore experiments the range is $[0,8]$. The upper bound is informed by a side-experiment discussed in Section 4.2.
>
> "instead of choosing $k_1$ and $k_2$ at hand, could we use a bandit algorithm for selecting premises? Here, the “context/user” would be the theorem to prove, and the “items/documents” would be the candidate premises."
> - We would like to note that even after picking $k_1$ or $k_2$, the hard part really is to find which $k_1$ or $k_2$ sized subset of premises to pick out of 20000 available premises. So we are guessing under this suggestion, that there is ranking already available? If so, we agree with the reviewer that it might be possible to further improve the results by fine-tuning this as well other hyperparameters picked from a range. That said, our focus in this work was to come up with a ranking in the first place, which is the hard part. As we mentioned above, we did not tune as it would be computationally prohibitive.  So rather than picking a fixed value in a experiment, we are making attempts over the course of a single experiment with different values to add more diversity.
>
> "it would be nice to give more details about the (higher-order) language used to define expressions"
> - Thanks for the suggestion, we will add more detail and update the rebuttal revision.

---

> > ### Comment · AnonReviewer3 · 2020-11-23
> > **Re: Re: AnonReviewer3**
> >
> > Thanks for your answers. Since the hyper-parameters $k_1$ and $k_2$ take an important role in this study, I would suggest adding a short paragraph or a footnote (in the experimental section of the main paper) that explains how they are generated.

---

> > > ### Author Response · Authors · 2020-11-23
> > > **Note on $k_1$ and $k_2$**
> > >
> > > Thanks for your suggestion. We have added a discussion on picking $k_1$ and $k_2$ in Section 5 (under a new paragraph titled "Hardware and hyperparameters").

---

> > > ### Author Response · Authors · 2020-11-23
> > > **HOL language description**
> > >
> > > Also -- forgot to mention in the previous comment, we have also added a brief description of the expressions as Appendix C, with additional pointer for more details on the foundations. Please take a look, and thanks again for your helpful suggestions.

---

### Official Review · AnonReviewer4 · 2020-10-29
**Simple idea, interesting empirical results.**

**Rating:** 6
**Confidence:** 4

**Review:**

The authors propose an RL approach to theorem proving focusing on improving the exploration aspect and demonstrates empirically that the proposed approach can perform well without requiring human proofs.

The paper is easy to read and the presentation is clear.

The authors address a very challenging issue of exploration in RL. Here, the proposed idea is to impose a preference on the action space being explored based on the tf-idf metric. The idea is itself not new but the way it is used in the RL setup, together with the presented empirical results, are relevant contributions.

With respect to the results in Figure 4, I wonder whether the authors can provide additional details in terms of the computational cost involved in each approach. In particular, what is the price to pay when going from pure-human to zero-explore? One would also imagine that with enough time resources, even zero-reference can eventually match zero-explore in validation score, but at what price?

It seems to me that the average proof length is a crucial parameter when it comes to scalability, since one would believe that the search complexity grows exponentially in the proof length. I assume that the use of tf-idf metric greatly reduced the "branching factor" in the initial search when the model is poor. Perhaps the author can comment on this so that the reader can judge whether a similar approach can be used in a different domain.

---

> ### Author Response · Authors · 2020-11-17
> **Re: AnonReviewer4**
>
> Thanks for your review, and thought-provoking comments and questions.
>
> * "the authors can provide additional details in terms of the computational cost involved in each approach" That is a very relevant question, thanks for raising it. We have updated Appendix B to address this question in detail. Briefly: the best zero loop takes 1.35 times the resources taken by the best human loop to reach 50% (continuous) validation performance. Looking purely locally at generating actions, the impact of adding tf-idf ranking doesn't add much relative to other components. Please take a look at Appendix B.3 let us know if it addresses your question.
>
> * "with enough time resources, even zero-reference can eventually match zero-explore" -- theoretically, yes, but practically, we think it will take several orders of magnitude more resources.
>
> * On for final question/comment. We looked at the average proof length statistic (of theorems that were proven on continuous validation at each round), and do not see a clear trend/difference there (across rounds, or human-vs-non human). That said, we think your intuition about "branching factor" is probably correct in the sense that there is already a very large "branching factor" for premise selection at the first step itself for tactic parameters. For tactics that take theorem parameters, the premise selection is picking a theorem from a knowledge base of up to 20,000 theorems.  For tactics that take theorem parameter lists, it is a subset of theorem parameters, if we restrict to 20 theorems, there is a really large number of possible actions (20000 choose 20). Would love to hear/discuss other application scenarios you have in mind.

---

### Official Review · AnonReviewer1 · 2020-10-29
**Learning to Reason in Large Theories without Imitation**

**Rating:** 6
**Confidence:** 4

**Review:**

In this paper, they tackle the challenge of learning to automated theorem (ATM) proving without any human imitation. Specifically the problem they focus on is premise selection. They find that while a vanilla RL scheme for ATM without any imitation learning of existing proofs frequently gets stuck and is not able to prove very many theorems, when a portion of the premises are selected via a simple term frequency-inverse document frequency (tf-idf) rule,  it dramatically increases the fraction of theorems proved — approaching the fraction of theorems proved with imitation learning of existing proofs.

In general I like this idea. While the approach is simple, the authors convincingly demonstrate that it is effective and it’s certainly interesting to learn that it is effective in this novel and highly important context. The authors also do several ablation experiments to thoroughly evaluate the components of the system.

A couple of questions —
(1) Would taking into account the similarity of the newly generated sub-goals as per tf-idf (or some other metric) with respect to past sub-goals that had short proofs help?
(2) Is the allowed maximum runtime the same for the systems compared with and the new system?

---

> ### Author Response · Authors · 2020-11-16
> **Re: AnonReviewer1**
>
> Thanks for the review, and your helpful comments and questions.
>
> (1) This is an interesting idea, though we are not sure how much it would help. Even when subgoals are syntactically similar, their proofs can be very different. Thus utilizing as-is short proof of H given a newly generated G (where G and H are similar) might not help much. In our experience, syntactic similarity (tf-idf-based or otherwise) is inferior to the semantic representations that deep neural networks can capture when given enough training data ([Lee et al., 2020] is an excellent example). In our work, tf-idf is a mechanism for bootstrapping and RL exploration, in order to train a deep neural network (graph neural networks in our case) for premise selection. The deep neural network becomes the dominant force, and does the heavy lifting.
>
> (2) Yes, the numbers are comparable as mentioned in Appendix A: "To keep this work comparable to prior results on the HOList benchmark, we have verified that the timeout and maximum explored nodes are the same as in Paliwal et al. [2020]. We have also verified that the results in Bansal et al. [2019] are within half a percent of the reported results with these timeout and maximum explored nodes.". We have also provided a full set of evaluation hyperparameters.
>
> References:
> * Dennis Lee, Christian Szegedy, Markus Rabe, Sarah Loos, Kshitij Bansal. Mathematical Reasoning in Latent Space. International Conference on Learning Representations (ICLR), 2020
> * Aditya Paliwal, Sarah Loos, Markus Rabe, Kshitij Bansal, and Christian Szegedy. Graph representations for higher-order logic and theorem proving. AAAI, 2020.
> * Kshitij Bansal, Sarah M Loos, Markus N Rabe, Christian Szegedy, and Stewart Wilcox. HOList: An environment for machine learning of higher-order theorem proving. International Conference on Machine Learning, ICML 2019.

---

### Official Review · AnonReviewer2 · 2020-10-30
**Official Blind Review #2**

**Rating:** 6
**Confidence:** 3

**Review:**

This paper presents how to train a theorem prover without access to human proofs using deep reinforcement learning. The paper views premise selection as an information retrieval problem where the goal to prove is the query and the knowledge base of previously proven theorems is the document set. The proposed method achieves better theorem proving performance than systems purely trained on human proofs, and approaches the performance of a prover trained by a combination of imitation and reinforcement learning. Overall, the paper is well organized and easy to follow.

Reasons to accept the paper:
1. The paper demonstrates that training the theorem prover without human data can succeed when using deep reinforcement learning.
2. The paper provides a side-by-side comparison of the effect of the availability of human proofs on the final theorem proving performance.
3. Experimental results show that the theorem prover trained without human proofs outperforms provers that are trained only on human proofs. Also, multiple ablation studies are provided to understand the underlying properties of the proof assistant and reinforcement learning setup.

Reasons to reject the paper:
1. The computational complexity and cost of the proposed method is not discussed in detail. Since the method is proposed to reason in large theories, the training efficiency should be important.
2. The proposed method relies on tf-idf based premises selection, which may limit the application scenario of the proposed method.

---

> ### Author Response · Authors · 2020-11-17
> **Re: AnonReviewer2**
>
> Thank you for your review, and we greatly appreciate your comments to help us improve the paper.
>
> Regarding the comment on "computational complexity" -- you raise an important question, and though we discussed this briefly in Appendix, based on your and another reviewer's comment realized that this could be expanded upon. Accordingly, we have expanded Appendix B. To summarize: looking locally at generating actions, the impact of adding tf-idf ranking doesn't add much relative to other components. Looking at it overall, the best zero loop takes 1.35 times the resources taken by the best human loop to reach 50% (continuous) validation performance. Of course, both of these use a lot of CPU resources, which a pure imitation approach doesn't. Please take a look at Appendix B.3, and let us know if it doesn't address your question.

---

### Official Review · AnonReviewer5 · 2020-11-06
**Review for "Learning to Reason in Large Theories without Imitation"**

**Rating:** 4
**Confidence:** 4

**Review:**

Summary:

The authors apply reinforcement learning to automated theorem proving to eliminate the need for human-written proofs as training data. During training, the prover is improved incrementally by using the current version to prove theorems and add proved theorems to the training data. The authors propose to use TF-IDF for premise selection. They show that it enables the prover to outperform a supervised learning baseline without using human proofs.


Strengths:

The paper addresses an interesting and important problem—learning theorem provers without explicit supervision from human proofs.

The evaluation is sound. The authors are able to show some benefits from their IF-IDF premise selection. The proposed method outperforms the baseline on the HOList benchmark.

The paper is well-written overall, though some sentences are difficult to parse, e.g.,  "comparison OF the effect OF availability OF human proofs" in the intro.

Weaknesses:

The method is not new. There exists a body of work leveraging reinforcement learning to learn theorem provers [A, B, C, etc.]. Many of them also do not rely on human proofs, e.g., [B]. Although the authors may be the first to apply reinforcement learning to prove theorems in interactive theorem proving environments, this is not a sufficiently novel contribution.

[A] Kaliszyk, Cezary, et al. "Reinforcement learning of theorem proving." Advances in Neural Information Processing Systems. 2018.
[B] Piotrowski, Bartosz, and Josef Urban. "ATPboost: Learning premise selection in binary setting with ATP feedback." International Joint Conference on Automated Reasoning. Springer, Cham, 2018.
[C] Zombori, Zsolt, Josef Urban, and Chad E. Brown. "Prolog technology reinforcement learning prover." arXiv preprint arXiv:2004.06997 (2020).


Besides, using TF-IDF for premise selection is not new. In the related work section, the authors say, "Gauthier et al. (2017) uses a tf-idf based premise selection model, but does not learn a model." I do not understand what the authors mean by "does not learn a model." It would be great if the authors can further clarify the difference with Gauthier et al.

---

> ### Author Response · Authors · 2020-11-13
> **Re: AnonReviewer5**
>
> Thank you for your comments and questions.
>
> We compare against [A;B;C] in Section 2, and as we note there none of [A;B;C] use deep neural networks, which have demonstrated the current state-of-the-art results in this domain. [A;C] are experiments for RL in proof search, but do not address the issue of premise selection from a large knowledge base in any way.
>
> We are also happy to elaborate on the difference with Gauthier et al. (2017). In our work, tf-idf is a mechanism for bootstrapping and RL exploration, in order to train a deep neural network (graph neural networks in our case) for premise selection. In the final evaluation (validation performance), only the deep neural network itself is used for premise selection. On the other hand, Gauthier et al. (2017) rely on a static tf-idf based similarity metric alone for premise selection.
>
> This is important since deep neural networks (*if there is enough training data*) are superior [Alemi et al 2016, Wang et al. 2017, Paliwal et al. 2020] for premise selection from large knowledge bases. This is because they are able to learn more semantic representations, not relying on purely syntactic aspects such as what terms or operators appear. This is the reason most recents works for premise selection use deep neural networks. The difference between use of only tf-idf to deep neural networks on the HOList benchmark is also demonstrated in our paper (Section 6, Ablation Studies).
>
> These state-of-the-art approaches for premise selection have so far relied on human proof data. We remove this reliance with a novel learning system, inspired from ideas that were only used separately in previous works. Our evaluation and ablation studies (see Sections 5 and 6) establish that the separate pieces alone do not suffice. Also, we provide a side-by-side comparison of the final theorem proving performance in the presence and absence of human proofs -- something that has been lacking in the literature.
>
> References
> - Thibault Gauthier, Cezary Kaliszyk, and Josef Urban. TacticToe: Learning to reason with HOL4 tactics. International Conference on Logic for Programming, Artificial Intelligence and Reasoning, Maun, 2017.
> - Alexander A. Alemi, François Chollet, Niklas Eén, Geoffrey Irving, Christian Szegedy, and Josef Urban. Deepmath - deep sequence models for premise selection. Advances in Neural Information Processing Systems, 2016.
> - Mingzhe Wang, Yihe Tang, Jian Wang, and Jia Deng. Premise selection for theorem proving by deep graph embedding. Advances in Neural Information Processing Systems, 2017.
> - Aditya Paliwal, Sarah Loos, Markus Rabe, Kshitij Bansal, and Christian Szegedy. Graph representations for higher-order logic and theorem proving. AAAI, 2020.

---

> > ### Public Comment · ~Bartosz_Piotrowski1 · 2020-11-16
> > **Utilizing human expertise for training premise selection in previous works**
> >
> > Many thanks to the authors for the interesting work.
> >
> > I wanted to clarify the issue of using human expertise for training premise selection in the previous works. In [B] there are 3 experimental scenarios in which premise selection is performed. The first two of them, indeed, do use human-originating training examples. However, in the last scenario, which is the main experiment of the paper, we start with no training proofs at all (neither human-originating nor purely ATP-originating). When the learning-proving feedback loop is started, the prover makes an initial pass using standard settings and a low time limit, and attempts to find proofs using all the premises allowed by the chronology in the formal library. Then the loop bootstraps from the few simple proofs found in this way; the premise selection model may be trained to provide the advice, more proofs is found, the model becomes stronger, etc.
> >
> > The scenario of starting with zero training proofs was used in a (much) earlier work too [D]. (There, the setting is actually more sophisticated.)
> >
> > [D] Josef Urban et al., MaLARea SG1 -- Machine Learner for Automated Reasoning with Semantic Guidance. IJCAR 2008

---

> > > ### Author Response · Authors · 2020-11-16
> > > **Corrected**
> > >
> > > Thank you for your clarification. I apologize for the misleading comparison in my comment, I have edited the comment to remove that. Though the comparison in the paper itself was accurate, I somehow missed that when rereading the paper for drafting the reply.

---

> > > > ### Public Comment · ~Bartosz_Piotrowski1 · 2020-11-16
> > > > **Re: Corrected**
> > > >
> > > > Thank you!

---

> > ### Comment · AnonReviewer5 · 2020-11-24
> > **Followup questions**
> >
> > Thanks to the authors for their clarification.
> >
> > ## Comparisons with Prior Work in RL + Theorem Proving
> >
> > Indeed, none of [A, B, C] used deep neural networks. But I don't think this is a valid difference between this paper and prior work. [A, B] discussed or experimented with deep neural networks. They didn't use it in their final methods because they found XGBoost is faster and equally accurate in their task.
> >
> > Though many theorem proving in ITP papers use DNNs (Bansal et al., Paliwal et al. Yang and Deng, etc.), they didn't compare with non-DNN method, and it is unclear whether DNNs have a performance advantage in the theorem proving domain.
> >
> > Furthermore, the use of DNNs is not a contribution of this paper. The authors are using exactly the same model architecture as Paliwal et al.
> >
> >
> > Bartosz raised a good point, which is also acknowledged by the authors, that [B] also performs premise selection in large knowledge bases without human supervision.
> >
> >
> >
> > ## The Use TF-IDF
> >
> > The authors mentioned that their method uses TF-IDF only during training but not during evaluation. Is this a significant design choice? What will happen if we also include premises selected by TF-IDF during evaluation?
> >
> >
> > Using TF-IDF for premise selection seems to be simplistic. It essentially selects premises that are similar to the current goal in a very shallow sense (treating both as bags of tokens).  Conceptually, the right premise does not need to be similar to the goal. Since TF-IDF works empirically, I'm wondering if authors have tried alternative methods for exploration in premise selection. If TF-IDF works, I would guess there is a large space of methods that are at least as good as TF-IDF.
> >
> > Related to the previous question, maybe the right premise is frequently similar to the goal in HOL Light. But I'm not sure if that's universally true for other proof systems.
> >
> > What is the vocabulary for TF-IDF? How large is it?
> >
> >
> >
> > ## Additional Questions
> >
> > While addressing theorem proving, the paper actually focuses on premise selection. How important is premise selection compared to selecting the right tactic? Maybe it would help to give an estimate of the number of tactics and the number of potential premises.  Also, different proof assistants may have a different number of tactics. Maybe HOL Light has only a handful of simple tactics, while its expressiveness relies heavily on premises. There may be other proof assistants with more complex designs in the space of tactics.  If that's the case, the proposed method may not generalize beyond HOL Light.
> >
> > Are there tactics taking more than one premises? And how is it handled?
> >
> > In the introduction, the authors state, "We establish the underlying properties of the proof assistant ... that makes our approach work." Maybe I have missed it, but I didn't see any experiments about the underlying properties of the proof assistant.
> >
> >
> > In the bootstrapping ablation: "we try to understand to what extent is the failure a bootstrapping issue". What do the authors mean by "bootstrapping issue"?
> >
> >
> >
> > ## Additional Comments
> >
> > It would be great to use bold font in Table 1 to mark the best method.
> >
> > If the dropout doesn't bring any statistically significant benefit, it would be better to remove it from the paper.

---

> > > ### Author Response · Authors · 2020-11-25
> > > **Quick response**
> > >
> > > [EDIT: This is a quick response, we will try to also answer the remaining questions soon in more detail.]
> > >
> > > "valid difference between this paper and prior work", "the use of DNNs is not a contribution of this paper" etc. -- the methods which use DNN for premise selection rely on human proof data. The ones which don't use DNN don't get state of the art performance. This is also established in our ablation experiments. The ones using only tf-idf for premise selection do significantly worse. So those existing methods don't seem to extend. Also, we have found continuous exploration with additional premise mixed helps quite a bit, which is also different from prior works.
> > >
> > > "What will happen if we also include premises selected by TF-IDF during evaluation" -- If only tf-idf is used during training is answered in ablation experiments. We believe tf-idf during evaluation wouldn't add much, since the deep neural network is much more powerful.
> > >
> > > "Since TF-IDF works empirically, I'm wondering if authors have tried alternative methods for exploration in premise selection. If TF-IDF works, I would guess there is a large space of methods that are at least as good as TF-IDF." -- our idea was to use a minimally engineered solution. There might be other possibilities, but our experiments show this allows us to exceed pure human imitation and  reach >90% of performance of human RL loop.
> > >
> > > "Additional Questions" -- most of these questions are already answered in the paper
> > >
> > > There are on avg over 10,000 premises and 41 tactics in the HOList benchmark.
> > >
> > > "Maybe I have missed it, but I didn't see any experiments about the underlying properties of the proof assistant." -- see section 4.2/4.3
> > >
> > > "we try to understand to what extent is the failure a bootstrapping issue" -- this is explained in section 5, but just in case -- to what extent it is just a lack of initial training data; and compare that to how much the additional continuous exploration helps.

---

> > > > ### Author Response · Authors · 2020-11-25
> > > > **Answer to remaining followup questions**
> > > >
> > > > "maybe the right premise is frequently similar to the goal in HOL Light" -- it is important to note that our approach doesn't rely on mere syntactic similarity. From our ablation experiments we conclude that would not give great results. As the model trains further the GNN model does more heavy lifting. The tf-idf adds additional premises to explore with the idea that, if they work, they will get added to the training data for the GNN model.
> > > >
> > > > "other proof systems" -- hopefully our experiments, esp Section 4 establishing the underlying conditions, throw important light on where such an approach (or a modification of it) might work.
> > > >
> > > > "What is the vocabulary for TF-IDF? How large is it?" -- Vocabulary is the set of tokens in the benchmark. The size of vocabulary is 885. We have also added this in the paper.
> > > >
> > > > "How important is premise selection compared to selecting the right tactic? Maybe it would help to give an estimate of the number of tactics and the number of potential premises." -- There are a total of 16623 theorems in the benchmark we evaluated on. When proving the i-th statement, the preceding (i-1) theorems are available as premises. There are 41 tactics. So it would be fair to say that premise selection is really the hard aspect (and also thus has been looked at by several works in the past, across various provers).
> > > >
> > > > "Are there tactics taking more than one premises? And how is it handled?" -- Yes, there are tactics that takes premise lists, guessing that is what your question was trying to ask. They are handled by selecting the top-k ranked premises being passed as input. Examples of these would be REWRITE and MESON, which are also discussed in Section 4.2.
> > > >
> > > > "It would be great to use bold font in Table 1 to mark the best method." -- Thanks, we have added that.

---

### Decision · Program_Chairs · 2021-01-07
**Final Decision**

**Decision:**

Reject

**Comment:**

*Overview* This paper applies RL to automated theorem proving to eliminate the need for human-written proofs as training data. The method uses TF-IDF for premise selections. The experiments compared with supervised baseline demonstrate some good performance.

*Pro* The paper provides a side-by-side comparison of the effect of the availability of human proofs on the final theorem proving.
The experiments compared with supervised baseline show that the proposed method has good performance even without human knowledge. The prosed TF-IDF selection algorithm addresses a challenging issue in exploration of RL.

*Con* The reviewers primarily concern about  the novelty of the methods. It appears the method is not new since there exist a body of work leveraging RL to learn theorem provers. The tasks are also not novel.  After rebuttal, the reviewers are not convinced that the novelty is significant enough for ICLR. The reviewers are also concerned that the proposed method might not be easily generalized to other tasks.

*Recommendation* Although the proposed method and experiment demonstrate some merits, there is a lack of novelty in terms of approaches. Since existing results already consider similar methods and similar tasks, it would make the paper stronger if thorough experimental comparisons are performed.